# The Effect of Endurance and Endurance-Strength Training on Bone Health and Body Composition in Centrally Obese Women—A Randomised Pilot Trial

**DOI:** 10.3390/healthcare10050821

**Published:** 2022-04-28

**Authors:** Małgorzata Jamka, Sylwia E. Piotrowska-Brudnicka, Joanna Karolkiewicz, Damian Skrypnik, Paweł Bogdański, Judyta Cielecka-Piontek, Gulnara Sultanova, Jarosław Walkowiak, Edyta Mądry

**Affiliations:** 1Department of Pediatric Gastroenterology and Metabolic Diseases, Poznan University of Medical Sciences, Szpitalna Str. 27/33, 60-572 Poznań, Poland; mjamka@ump.edu.pl (M.J.); jarwalk@ump.edu.pl (J.W.); 2Department of Physiology, Poznan University of Medical Sciences, Święcickiego Str. 6, 61-781 Poznań, Poland; sylwia.piotrowskaa@gmail.com; 3Department of Clinical Biomechanics and Physiotherapy in Motor System Disorders, Faculty of Health Science, Wroclaw Medical University, Grunwaldzka Str. 2, 50-355 Wrocław, Poland; 4Department of Food and Nutrition, Poznan University of Physical Education, Królowej Jadwigi Str. 27/39, 61-871 Poznań, Poland; karolkiewicz@awf.poznan.pl; 5Department of Treatment of Obesity, Metabolic Disorders and Clinical Dietetics, Poznan University of Medical Sciences, Szamarzewskiego Str. 82, 60-569 Poznań, Poland; dskrypnik@ump.edu.pl (D.S.); pbogdanski@ump.edu.pl (P.B.); 6Department of Pharmacognosy, Poznan University of Medical Sciences, Rokietnicka 3, 60-806 Poznań, Poland; jpiontek@ump.edu.pl; 7West Kazakhstan Marat Ospanov Medical University, Maresyev Str. 68, Aktobe 030019, Kazakhstan; stomfak.zkgmu@mail.ru

**Keywords:** body composition, bone mineral content, bone mineral density, densitometry, exercise, obesity

## Abstract

There is no consensus exercise programme to reduce body weight and improve body composition simultaneously preventing bone loss or stimulating osteogenesis. This pilot study compared the effect of endurance and endurance-strength training on body composition and bone metabolism in centrally obese women. Recruited subjects were randomly assigned to three-month endurance (*n* = 22) or endurance-strength training (*n* = 22). Body composition, bone mineral density (BMD) and content (BMC) were assessed before and after the intervention and markers of bone formation and resorption were measured. Both training significantly decreased fat mass; however, endurance-strength training had a more favourable effect on lean mass for the gynoid area (*p* = 0.0211) and legs (*p* = 0.0381). Endurance training significantly decreased total body BMC and BMD (*p* = 0.0440 and *p* = 0.0300), whereas endurance-strength training only reduced BMD (*p* = 0.0063). Changes in densitometric parameters did not differ between the groups but endurance training increased osteocalcin levels (*p* = 0.04845), while endurance-strength training increased tartrate-resistant acid phosphatase 5b concentrations (*p* = 0.00145). In conclusion, both training programmes were effective in the reduction of fat mass simultaneously negatively affecting bone health. However, endurance-strength training seemed to be more effective in increasing lean mass. The study protocol was registered in the ClinicalTrials.gov database under the number NCT03444207, date of registration: 23 February 2018 (retrospective registration).

## 1. Introduction

Obesity is a major global health problem that is increasing in prevalence worldwide [1] and has a high impact on mortality and morbidity [2]. The prevalence of obesity globally has doubled in the last 30 years and currently, more than 13% of subjects are obese [1]. The World Health Organization defines obesity as a body mass index (BMI) higher than or equal to 30 kg/m^2^ [3]. An alternative definition based on abnormal or excessive fat accumulation is not well-established [4]. Obesity can be classified as peripheral and central (abdominal) according to fat distribution, with central obesity defined as a waist circumference greater than or equal to 94 cm in men and 80 cm in women for the European population [5]. Recently, central obesity has been considered to be a more important independent risk factor for several diseases [6,7], with visceral fat contributing to bone loss [8,9]. It has been shown that bone mineral density (BMD) decreases with an increase in the waist-to-hip ratio, which is an index of central obesity [10]. 

Physical activity is an effective method of prevention and treatment of many diseases, including obesity [11]. Furthermore, exercises may also have a favourable effect on bone health by increasing bone mass [12]. Several meta-analyses confirmed a significant effect of exercise on densitometric parameters; however, this depended on sex, age of the study population as well as the type and intensity of activity [12,13,14,15,16,17]. Considering the large variety of training programmes, there is also no doubt that some exercises may demonstrate favourable while others have negative effects on bone health. Moreover, there is no consensus exercise programme for bone mass [12]. A recent meta-analysis reported insufficient data to advocate a specific type of training for maintaining bone mass in obese subjects [18]. On the other hand, in a systematic review, Pinheiro et al. [19] reported that in subjects aged ≥ 65 years, only training programmes that were undertaken for at least 60 min and performed 2–3 times per week for at least seven months were effective in enhancing bone health. According to the general physical activity recommendation for adults, at least 150 min of moderate-intensity training per week should be performed to maintain good health. Training should be conducted at least three times per week and include both endurance and strength exercises [20,21].

Therefore, this pilot study compared the effect of two types of training programmes (endurance and endurance-strength) on body composition and bone metabolism in women with central obesity.

## 2. Materials and Methods

### 2.1. Study Design

The study was designed as a prospective randomised trial and adhered to the standards laid down in the Declaration of Helsinki. The study research protocol was approved by the Ethics Committee at the Poznan University of Medical Sciences in Poland (ref. 1077/12 and 753/13) and the research was conducted according to the consolidated standards of reporting trials guidelines for randomised parallel trials (see Appendix A) [22]. The study protocol was registered in the ClinicalTrials.gov database (NCT03444207) and can be found at: https://clinicaltrials.gov/ct2/show/NCT03444207 (accessed on 1 March 2022). Potential participants were provided with study information before enrolment and provided written informed consent before beginning any intervention. The study was conducted between January and March 2013.

### 2.2. Inclusion and Exclusion Criteria

Participants were recruited from the Department of Internal Medicine, Metabolic Disorders, and Hypertension, Poznan University of Medical Sciences in Poland. The inclusion criteria were as follows: women aged 18–65 years with central obesity defined as a BMI ≥ 30 kg/m^2^, waist circumference ≥ 80 cm, ≥33% body fat (assessed by bioelectrical impedance analysis), sedentary activity and stable body weight in the month before the trial (permissible deviation was ±1 kg). The exclusion criteria included: secondary obesity, secondary hypertension or non-controlled hypertension (blood pressure ≥ 140/90 mmHg), the necessity to modify antihypertensive or hypolipidemic treatment within three months before the trial, diabetes mellitus, cardiovascular diseases (coronary artery disease, stroke, transient ischaemic attack, congestive heart failure, or arrhythmia), acute or chronic inflammatory diseases, connective tissue diseases, arthritis, serious and uncontrolled kidney, hepatic or thyroid diseases, cancer diseases, pregnancy, breastfeeding, any addictions or abuse, drugs potentially affecting bone metabolism, dietary supplement intake and hormone replacement therapy. Moreover, participants had to complete at least 29 of 36 training sessions (around 80%) to be included in the analysis. 

### 2.3. Outcomes

This manuscript presents the results from the pilot study of a large project aiming to compare the effect of endurance and endurance-strength training on endothelial function (acronym: ENDOFIT) [23,24,25,26]. The primary outcome of the pilot study was the effect of endurance and endurance-strength training programmes on inflammatory markers, while the secondary outcomes included the impact of the training programmes on anthropometric parameters and body composition [27,28,29,30]. 

In the present study, we assessed the effect of exercises on densitometric bone parameters and bone turnover markers as well as on body composition. 

### 2.4. Interventions Protocol

Training for the endurance and endurance-strength groups consisted of three weekly sessions lasting 60 min performed on non-consecutive days (Monday, Wednesday and Friday) for 12 weeks. Both training programmes had equivalent volumes and differed in the nature of the effort. A certified trainer designed the training. All training sessions included three phases: warm-up phase, main phase and cool-down phase. The warm-up phase consisted of low intensity (50–60% of maximal heart rate (HRmax)) exercises performed for five minutes. The main phase lasted around 45 min and included cycling on an ergometer (Schwinn Evolution, Schwinn Bicycle Company, Boulder, CO, USA) with an individualised load (between 50–80% of HRmax) for the endurance group. In the endurance-strength group, the main phase consisted of an endurance component (25 min of cycling similar to the endurance group) and a strength component (20 min of strength exercises). The strength component was variable and repeated regularly every week to allow muscle power to regenerate as follows: on Monday–upper limb exercises with the barbell (weight: 1.6 kg) and two weights (weight: 1.25 kg), on Wednesday–spine-stabilising exercises, deep muscle-forming exercises and balance-adjusting exercises using a gym ball, on Friday–lower limb exercises with the use of a barbell. The number of repetitions of each exercise in the series was dependent on the subjects’ muscle strength and equal to the number of repetitions performed correctly. Participants performed three to six sets of exercises, and the goal number of repetitions was 16 in barbell curls and 30 in barbell squats. The resistance exercise load was 50–60% of the one-repetition maximum (RM). A detailed scheme of strength exercises is presented in Appendix A. The cool-down phase in both groups involved five minutes of cycling without load and five minutes of low-intensity stretching and breathing exercises. For each subject, the intensity of training was individually selected and was constant during all intervention periods. The exercises were supervised by a qualified and certified fitness instructor and medical rescuer or physician at a professional sports club, the Sports Club City Zen in Poznań, Poland. During the intervention, subjects were instructed not to change their dietary habits or daily activity. 

### 2.5. Anthropometric Parameters

Anthropometric measurements were performed in the morning, with light clothing and without shoes. Body weight and height were measured by a medical scale with a stadiometer to the nearest 0.1 kg and 0.5 cm, respectively. Waist circumference was measured using a flexible, inextensible tape with an accuracy of 0.5 cm. The BMI was calculated based on weight and height using the standard formula. Central obesity was defined as BMI ≥ 30 kg/m^2^ [3], waist circumference ≥ 80 cm [5] and percentage of body fat ≥ 33% [4].

### 2.6. Body Composition and Densitometric Parameters

During recruitment, body composition was evaluated to check if potential participants fulfilled the inclusion criteria by bioelectrical impedance analysis using the InBody 370 analyser (InBody Co. Ltd., Seoul, Korea). 

Before and after the intervention, body composition (fat mass and lean mass) and densitometric parameters (BMD and bone mineral content (BMC)) were assessed using dual-energy X-ray absorptiometry methods (General Electric Healthcare Lunar Prodigy Advance Medical Systems, Milan, Italy). BMD and BMC were measured at the lumbar spine (L1-L4), femoral neck and total body. Body composition analysis included the examination of the fat mass and lean mass for the total body, as well as arms, trunk, legs, male (android) and female (gynoid) areas. Analysis was performed using a standard scan mode (for moderately obese subjects) or thick scan mode (for extremely obese subjects), with an absorbed dose of radiation of 0.4 µGy and 0.8 µGy, respectively. The intra- and inter-individual coefficient of variation was less than 1% for bone mass, 2.2% for fat mass and 1.1% for lean mass. The measurement lasted about 15 min, with participants fully informed about the objectives and procedures of the analysis. Participants wore light clothing and removed all metal objects and were instructed not to exercise for 24 h before testing. The same technician performed and assessed all scans, with densitometer reset daily as per the manufacturer’s recommendations to assure quality. Measurements were performed at the Clinical Densitometry Laboratory at the Hetmańska Centre in Poznań, Poland.

### 2.7. Physical Capacity

Physical capacity was measured by the graded exercise test using a cycling ergometer (Kettler^®^ DX1 Pro, Kettler, Ense, Germany) and an Oxycon mobile device (Viasys Healthcare, Hochberg, Germany). Before each test, the system was calibrated according to the manufacturer’s instructions. The initial load was 25 W and the load increased every two minutes to record the ventilation rate, carbon dioxide production, oxygen consumption and respiratory rate. A ventilatory threshold was assessed using the V-slope method, with each test lasting from four to 15 min, depending on the participant’s fitness status. The assessment was conducted at the Department of Physiology, Poznan University of Physical Education, Poznań, Poland and the detailed physical capacity procedures have been described previously [28].

### 2.8. Circulatory System Measurements

Blood pressure and heart rate were measured with a digital blood pressure monitor at rest and during the graded exercise test blood pressure and heart rate were measured with a digital blood pressure monitor (Model 705IT, Omron Corporation, Kyoto, Japan). Resting measurements were performed according to the European Society of Hypertension recommendation [31]. The heart rate during training was monitored with a Suunto Fitness Solution device (Suunto, Vantaa, Finland) to monitor the exercise intensity in real-time. The current heart rate and percentage of HRmax were recorded for each participant and the results were compared to the parameters achieved during the graded exercise test. Considering the sedentary activity of our participants and the risk of injury, a one-RM was calculated using the Brzycki formula [32], frequently used in other studies [33,34,35,36].

### 2.9. Biochemical Markers 

Fasting blood samples were collected in the morning after a 12 h break from the last training session and meal. Blood samples were centrifuged and stored at −80 °C until analysis. The following markers of bone formation were measured before and after the intervention period: osteocalcin (OC, MicroVue™ Osteocalcin EIA kit, Quidel Corporation, San Diego, CA, USA), bone alkaline phosphatase (BAP, MicroVue™ BAP EIA kit, Quidel Corporation, San Diego, CA, USA) and bone resorption: tartrate-resistant acid phosphatase serum band 5 (TRAP 5b, MicroVue™ TRAP 5b EIA kit, Quidel Corporation, San Diego, CA, USA) and α-collagen type 1 cross-linked C-terminal telopeptide (CTX-1, Crosslaps^®^ CTX-1 ELISA kit, Immunodiagnostic Systems Holdings PLC, Boldon, Great Britain, UK). The serum levels of bone turnover markers were quantified by an enzyme-linked immunosorbent assay. All measurements were performed at the Department of Pediatric Gastroenterology and Metabolic Diseases, Poznan University of Medical Sciences, Poznań, Poland. 

### 2.10. Randomisation and Blinding

A computer-generated randomisation list was created and used to assign the study population into two groups: endurance and endurance-strength (allocation ratio: 1:1). Blocking randomisation was performed by an independent researcher, with the allocation sequence concealed until enrolment. Due to the nature of the effort, study participants and researchers were not blinded, so only outcomes assessors and statisticians were not aware of the allocation.

### 2.11. Minimum Sample Size

The minimum sample size was calculated using the Statistica 6 PL software (TIBCO Software Inc., Palo Alto, CA, USA), indicating that at least 16 subjects per group should be recruited to obtain 80% power (α = 0.05, β = 0.2).

### 2.12. Statistical Analysis

Data analysis was performed using the Statistica 12 PL software (TIBCO Software Inc., Palo Alto, CA, USA). Data are presented as mean and standard deviation (SD) with 95% confidence interval (95% CI) and median and interquartile range (IQR; Q1–Q3). The normal distribution of variables was verified with the Shapiro–Wilk test and the Fisher–Snedecor test was used to assess the equality of variances. Unpaired t-tests or the Mann–Whitney U-test for normally or non-normally distributed continuous variables were used to examine differences between the study groups. A paired t-test or the Wilcoxon test was used to analyse the intragroup change from baseline. Pearson linear correlations or Spearman’s correlation analysis were used to analyse the association between changes in densitometric variables, bone turnover markers and body composition. The ANCOVA test, adjusted for the baseline measures as a covariate, was used to compare the changes in each variable between groups. Non-normally distributed data were normalised before the analysis using a Box-Cox transformation, then the data were back-transformed for ease of interpretation of results. A *p*-value < 0.05 was considered statistically significant. 

## 3. Results

### 3.1. Study Flow

The study flow diagram is presented in Figure 1. In total, 163 obese females were screened at the Department of Internal Medicine, Metabolic Disorders, and Hypertension, Poznan University of Medical Sciences, Poznań, Poland, of which only 44 women met the inclusion criteria and were randomly assigned to the endurance (*n* = 22) and endurance-strength (*n* = 22) groups. Eventually, 38 subjects completed the study, 21 from the endurance group and 17 from the endurance-strength group. Six subjects were lost to follow-up due to poor compliance. There were no significant differences at baseline in the analysed variables between groups (see Table 1, Table 2, Table 3 and Table 4). The baseline characteristics of the study population were also published previously [27,28,29,30]. No adverse effects were noted.

### 3.2. The Effect of Physical Activity on Bone Turnover Markers

The impact of both training programmes on bone turnover parameters is shown in Table 2. After the 12-week intervention, endurance training significantly increased OC levels (*p* = 0.04845), while endurance-strength training significantly increased TRAP 5b (*p* = 0.00145). No effects of any other bone turnover markers were observed.

### 3.3. The Effect of Physical Activity on Densitometric Parameters

The effect of endurance and endurance-strength exercises on BMD and BMC is illustrated in Table 3. Endurance training significantly decreased BMC and BMD in the total body (*p* = 0.0440 and *p* = 0.0300), whereas endurance-strength training only reduced BMD (*p* = 0.0063). 

### 3.4. The Effect of Physical Activity on Body Composition

The effects of endurance and endurance-strength training programmes on body composition are presented in Table 4, showing that the 12-week endurance and endurance-strength training significantly reduced the fat mass (endurance training: total body: *p* < 0.0001, male (android): *p* < 0.0001, female (gynoid): *p* < 0.0001, leg: *p* = 0.0002 and trunk: *p* = 0.0011, and endurance-strength training: total body: *p* < 0.0001, male (android): *p* = 0.0003, female (gynoid): *p* < 0.0001, arms: *p* = 0.0442, leg: *p* = 0.0006 and trunk: *p* = 0.0014). Moreover, both training programmes significantly increased lean mass in the female (gynoid) parts (endurance group: *p* = 0.0013 and endurance-strength group: *p* = 0.0001) and legs (endurance group: *p* = 0.0129 and endurance-strength group: *p* = 0.0003). Endurance training also reduced the android (male) lean mass (*p* = 0.0117), while endurance-strength training increased the total body lean mass (*p* = 0.0005).

### 3.5. Comparison of the Effect of Endurance and Endurance-Strength Training on Body Composition, Bone Turnover and Densitometric Parameters

A comparison of the effect of both training programmes on the analysed variables is presented in Table 5, Table 6 and Table 7, with significant differences in the impact on lean mass. Endurance-strength training had a more favourable effect on lean mass for the gynoid (female) area (mean (95% CI): 227 (71–400) vs. 568 (325–836) g, *p* = 0.0211) and legs (mean (95% CI): 454 (45–886) vs. 1141 (674–1630) g, *p* = 0.0381). No differences between other parameters were found. 

### 3.6. Association between Changes in Densitometric Parameters, Bone Turnover Markers and Body Composition

In the total population, changes in BMD in the total body correlated positively with changes in total body lean mass (*r* = 0.5766, *p* = 0.0001) and trunk lean mass (*r* = 0.4418, *p* = 0.0055), while changes in BMC in the total body negatively correlated with changes in arm fat mass (*r* = −0.4212, *p* = 0.0084), trunk lean mass (*r* = −0.4210, *p* = 0.0058) and total lean mass (*r* = −0.4998, *p* = 0.0014), while positive correlations were noted for changes in trunk fat mass (*r* = 0.4751, *p* = 0.0026) and total fat mass (*r* = 0.4284, *p* = 0.0073). Moreover, changes in TRAP5b levels were oppositely associated with changes in leg lean mass (*r* = −0.3368, *p* = 0.0386).

## 4. Discussion

This pilot study demonstrated that both training programmes significantly reduced fat mass but endurance-strength training had a more favourable effect on increasing lean mass. Moreover, our results suggested that both training programmes negatively affected bone health. Endurance training significantly decreased BMC and BMD in the total body, whereas endurance-strength training only reduced BMD. However, there were no differences in densitometric parameters between groups.

Previously, a few studies compared the effect of endurance and endurance-strength training on densitometric parameters [24,37,38,39,40]. Villareal et al. [37] evaluated the effect of six-month endurance, strength and combined training in dieting obese older adults and found that BMD in the total hip did not change in the strength group but decreased in the endurance group and the mixed group, with a more negative effect in the endurance group. In the same study, BMD in the total body and the lumbar spine (L1-L4) did not change significantly in any of the groups. Rossi et al. [38] assessed the effect of endurance and endurance-strength training after 16 weeks of intervention in postmenopausal women and reported no differences in BMC and BMD between groups. Additionally, Stensvold et al. [39] found that endurance, strength and endurance-strength training did not affect BMC after 12 weeks of intervention in participants with metabolic syndrome. Furthermore, Campos et al. [40] evaluated the effect of 12-month endurance and endurance-strength training in young adolescents, reporting a decrease in BMC with no changes in BMD in the endurance group and an increase in BMC for the total body in the endurance-strength group. In a recent paper, we also compared the effect of 12 weeks of endurance and endurance-strength training in centrally obese women aged 50–60 years, reporting no differences between groups in the effect on BMD and BMC in the femoral neck and the total body. However, endurance training was more favourable in maintaining BMC at the L1-L4 [24]. 

Several factors can affect bone mass and explain the differences in obtained results between studies, for instance, age [41] and sex [42]. The mean age of women in our study was 51.3 ± 8.3 years in the endurance group and 48.2 ± 11.2 years in the endurance-strength group. In women, the physiological process of bone loss usually starts around 40 years of age [43] but during the perimenopausal period, there is a more intensive decrease in BMD [44,45]. Therefore, performing the same exercises in men and women [42] as well as in different age groups [41] may have opposite effects. However, the physiological reduction in bone mass with age may only partly explain the negative effect of our intervention on densitometric parameters. Another critical factor influencing bone health is hormonal status. In women, adipose tissue is the second most important organ for oestrogen production (after the ovaries), the hormones responsible for maintaining proper BMD [46]. Therefore, the significant loss of fatty tissue, which occurs over a short period, should undoubtedly be regarded as a factor deteriorating the bone state [47,48,49]. 

Mechanical stimuli strongly influence bone tissue metabolism and may inhibit bone loss with age [50]. Bone can be stimulated by the working muscles and by the loads generated by body weight [51,52]. It is well known that body weight reduction in a short time might negatively affect bone health [53,54]. In our study, we noted a significant fat mass reduction after the intervention, which could partly explain the obtained results. Moreover, we found that changes in BMD in the total body positively correlated with changes in total lean mass, while changes in BMC in the total body were negatively correlated with changes in total lean mass and positively correlated with changes in total fat mass.

The type of exercises and their volume may also determine the effect of physical activity on bone mass [12]. In our research, the volume of exercises in both groups was similar; therefore, we assume that the observed differences between groups were related to the type of performed exercises. Several studies have shown that cycling negatively affects bone mass [55,56], whereas strength training may promote bone formation and does not affect bone resorption processes [57,58]. Our results seem to partly confirm these findings. In the studies conducted by Villareal et al. [37] and Rossi et al. [38], participants in the endurance group mainly performed weight-bearing exercises, including walking, climbing stairs and the positive effect of such exercise on osteogenesis has been previously confirmed. Shanb and Youssef [59] showed a beneficial effect of weight-bearing exercises for elderly subjects with osteoporosis, while Martyn-St. James and Carroll [60] demonstrated that endurance weight-bearing exercises combined with strength training are most beneficial on bone tissue in the meta-analysis. It is well known that sports like jumping, dancing, volleyball, basketball or running have more favourable effects on bones than swimming or cycling [12], with Rector et al. [61] and Stewart and Hannan [62] showing that the BMD of runners was significantly higher than in cyclists. However, Chen et al. [63] reported that at the hip and tibia, runners had higher bone mass compared to cyclists but cyclists had better bone strength and larger bone size.

During cycling, there is no load on the upper limbs, possibly explaining no effect on fat and lean mass in the arms observed in the endurance group and a decrease of fat mass in the endurance-strength group, which performed upper limb exercises. Contrary to the upper limbs, the lower limbs perform intensive work while cycling [64], which probably prevented the loss of BMD and BMC at the femoral neck, simultaneously reducing fat mass and increasing lean mass in both study groups with significantly greater changes in lean mass in the endurance-strength group. However, we also previously compared the effect of endurance and endurance-strength training on body composition, observing a significant decrease in fat mass and increase in free fat mass in legs and arms in both groups with a significantly higher decrease in fat mass in legs in the endurance-strength group [26]. Furthermore, contrary to our results, Nichols and Rauh [65] observed reductions in the BMD of the femoral neck in older cycling adults, while Beshgetoor et al. [66] demonstrated a decrease in BMD in the L1-L4 in similar-aged cycling women. 

The duration, frequency and length of the training programme may also determine the training effectiveness [12,67,68]. Our intervention included moderate-intensity training programmes performed three times a week for 60 min which was in line with the Physical Activity Guidelines for Americans recommendations [69]. Moreover, the duration of the intervention was 12 weeks; however, the duration of dense and spongy bone tissue remodelling is around 17 and 28 weeks, respectively [70]. However, in the study conducted by Villareal et al. [37], the intervention period and session duration were longer, lasting 16 weeks and 75–90 min, respectively, with an observable, negative effect on densitometric parameters. Furthermore, Campos et al. [40] assessed the effect of 12-month endurance and endurance-strength training, reporting a decrease in BMC with no changes in BMD in the endurance group and an increase in BMC for the total body in the endurance-strength group. Extending the intervention period in our study could guarantee the completion of the full cycle of bone remodelling. However, maintaining the motivation to systematically participate in training for an additional 12 weeks in our study group of centrally obese women with sedentary activity could also be associated with an increased drop-out rate.

Changes in bone turnover marker concentrations correspond to changes in bone tissue [71,72]. Previous studies reported that strength training promotes bone formation without enhancing resorption processes [73,74]. However, Woitge et al. [75] observed that eight weeks of interval sprints (anaerobic exercise) training combined with weight lifting training increased osteogenesis and bone resorption markers. In the same study, four-week running training (aerobic exercise) reduced both markers of bone resorption and bone formation. Our analyses showed a significant increase in osteocalcin levels (bone formation marker) in the endurance group and TRAP 5b levels (bone resorption marker) in the endurance-strength group, with no significant differences between groups. 

Body weight strongly influences changes in bone turnover markers [76,77]. In obese adults, lower levels of osteosynthesis and osteoresorption markers are observed compared to subjects with normal body weight [78]. Moreover, body weight reduction contributes to an increase in bone turnover markers [76]. Furthermore, the weight loss in previously untrained subjects beginning exercise training may temporarily outweigh bone formation [79]; therefore, the obtained results in the endurance-strength group may be due to the changes in body composition.

This pilot study has some limitations including the small sample size, lack of information on the women’s menopausal status and a wide age range. Moreover, a three-month intervention may be too short to assess the effect of training programmes on bone health. However, the primary aim of this pilot study was to compare the effect of endurance and endurance-strength exercises on inflammatory status. Therefore, the duration of the intervention was chosen based on the effect on the primary outcomes. Extending the intervention period for additional weeks in our population could be difficult and associated with decreased motivation and an increased drop-out rate in relatively small groups observed. Additionally, the strength training has some weaknesses as RM was calculated from the Brzycki formula [32] and was not directly measured. Furthermore, some additional parameters such as parathyroid hormone, vitamin D and calcium levels should be assessed to investigate bone metabolism in depth. Moreover, our results cannot be generalised to other populations, as to minimise the influence of sex [42], only centrally obese women were included in the study using strict inclusion and exclusion criteria. However, using these criteria allowed the selection of a homogenous group of subjects. 

Notwithstanding, the present study is a rationally designed pilot randomised trial that investigated the effectiveness of different training modalities in women living with central obesity. The strengths of the study also include a high compliance ratio (around 85%) and the elimination of the influence of eating habits. The results of this pilot study suggested the need for changing the physical activity applied in our study. Therefore, in a subsequent trial, a new training programme with cycling in a sitting position mixed with cycling in a standing position was used. This modification prevented bone loss [24]. 

## 5. Conclusions

In conclusion, three months of endurance and endurance-strength training with a dominant component of traditional cycling may reduce densitometric bone parameters simultaneously decreasing fat mass in women with central obesity. However, endurance-strength training seems to be more effective in increasing lean mass than endurance training, but further studies are needed to confirm these findings.

## Figures and Tables

**Figure 1 healthcare-10-00821-f001:**
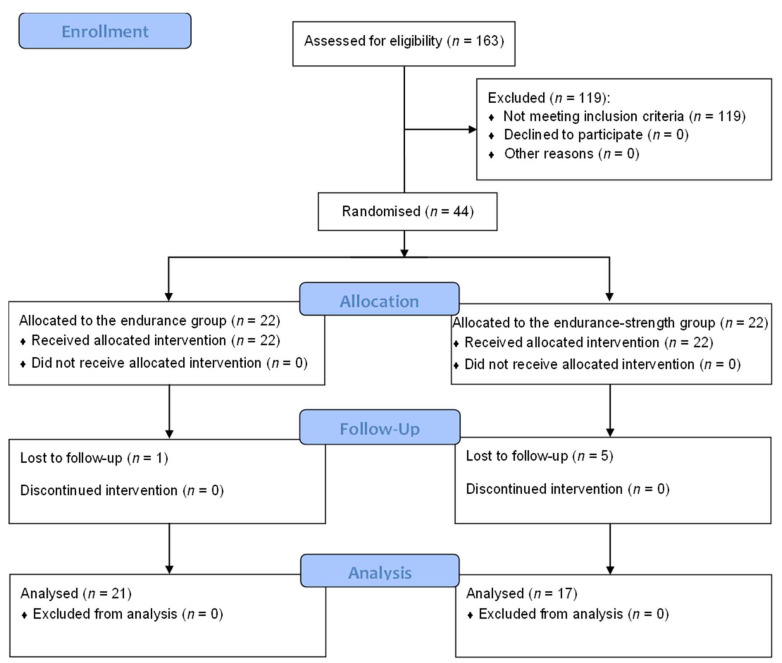
Study flow chart.

**Table 1 healthcare-10-00821-t001:** Baseline characteristics of the study population.

Variables	END Group (*n* = 21)		END–ST Group (*n* = 17)		*p* ^1^
	Mean ± SD (95% CI)	Median (IQR)	Mean ± SD (95% CI)	Median (IQR)	
Age [years]	51 ± 8 (48–55)	53 (43–58)	48 ± 11 (42–54)	49 (40–58)	0.3236
Weight [kg]	91.7 ± 11.8 (86.3–97.0)	90.2 (84.5–97.0)	94.5 ± 13.4 (87.6–101.4)	93.9 (80.1–101.3)	0.4928
Height [m]	1.61 ± 0.05 (1.59–1.64)	1.63 (1.58–1.65)	1.64 ± 0.06 (1.61–1.67)	1.66 (1.61–1.67)	0.1114
BMI [kg/m^2^]	35.17 ± 3.86 (33.41–36.93)	33.95 (32.38–37.51)	34.93 ± 3.82 (32.96–36.89)	33.75 (32.11–36.89)	0.8464
Waist circumference [cm]	110.8 ± 10.2 (106.1–115.4)	112.0 (102.0–115.0)	111.6 ± 11.3 (105.8–117.5)	109.5 (101.0–121.5)	0.8020
Fat [%] ^2^	46.9 ± 3.7 (45.2–48.6)	45.8 (44.5–49.3)	46.1 ± 5.1 (43.5–48.8)	46.2 (44.5–48.9)	0.5971

BMI–body mass index, END–endurance group, END-ST–endurance-strength group, IQR–interquartile range, SD–standard deviation, 95% CI–95% confidence interval, ^1^
*p*-value for differences between groups, and unpaired *t*-test, ^2^ assessed by bioelectrical impedance analysis.

**Table 2 healthcare-10-00821-t002:** The effect of training programmes on bone turnover marker levels.

Variables		END Group (*n* = 21)		END–ST Group (*n* = 17)		*p* ^1^
		Mean ± SD (95% CI)	Median (IQR)	Mean ± SD (95% CI)	Median (IQR)	
BAP [U/L]	PRE	25.52 ± 5.94 (22.81–28.22)	25.39 (20.36–29.7)	24.80 ± 8.22 (20.57–29.03)	23.58 (19.26–29.85)	0.7587 ^3^
	POST	26.60 ± 7.11 (23.37–29.84)	26.53 (22.01–30.01)	24.25 ± 7.78 (20.25–28.25)	23.74 (18.00–27.74)	0.3366 ^3^
*p* ^2^		0.4234 ^5^		0.5750 ^5^		
CTX–1 [ng/mL]	PRE	0.41 ± 0.19 (0.32–0.49)	0.40 (0.28–0.52)	0.42 ± 0.21 (0.32–0.53)	0.33 (0.30–0.52	0.8847 ^4^
	POST	0.41 ± 0.19 (0.33–0.50)	0.36 (0.28–0.55)	0.47 ± 0.22 (0.36–0.59)	0.44 (0.33–0.57)	0.3702 ^3^
*p* ^2^		0.8077 ^5^		0.2274 ^5^		
OC [ng/mL]	PRE	4.98 ± 2.34 (3.92–6.05)	5.37 (3.55–6.97)	5.77 ± 2.40 (4.53–7.00)	4.90 (4.12–7.64)	0.3178 ^3^
	POST	5.91 ± 2.21 (4.91–6.92)	6.48 (4.73–9.98)	6.79 ± 2.41 (5.53–8.05)	6.34 (4.82–9.15)	0.2523 ^3^
*p* ^2^		0.0484 ^5^		0.0733 ^6^		
TRAP 5b [U/L]	PRE	1.88 ± 0.75 (1.53–2.22)	1.82 (1.26–2.55)	1.60 ± 0.75 (1.21–1.98)	1.66 (1.01–2.09)	0.2641 ^3^
	POST	2.02 ± 0.94 (1.59–2.45)	2.06 (1.10–2.70)	2.22 ± 0.75 (1.83–2.60)	2.31 (1.60–2.88)	0.5020 ^3^
*p* ^2^		0.1808 ^6^		0.0014 ^5^		

BAP–bone alkaline phosphatase, CTX-1–α-collagen type 1 cross-linked C-terminal telopeptide, END–endurance group, END-ST–endurance-strength group, IQR–interquartile range, OC–osteocalcin, PRE–pre-intervention value, POST–post-intervention value, SD–standard deviation, TRAP 5b–tartrate-resistant acid phosphatase serum band 5, 95% CI–95% confidence interval, ^1^
*p*-value for differences between groups, ^2^
*p*-value for differences between pre- and postintervention values, ^3^ unpaired *t*-test, ^4^ Mann–Whitney U-test, ^5^ paired *t*-test and ^6^ Wilcoxon test.

**Table 3 healthcare-10-00821-t003:** The effect of training programmes on bone parameters.

Variables		END Group (*n* = 21)		END-ST Group (*n* = 17)		*p* ^1^
		Mean ± SD (95% CI)	Median (IQR)	Mean ± SD (95% CI)	Median (IQR)	
**Total body**						
BMD [g/cm^2^]	PRE	1.224 ± 0.084 (1.186–1.262)	1.226 (1.152–1.290)	1.258 ± 0.100 (1.207–1.310)	1.231 (1.208–1.296)	0.2542 ^3^
	POST	1.208 ± 0.098 (1.163–1.252)	1.199 (1.135–1.271)	1.241 ± 0.089 (1.195–1.286)	1.236 (1.183–1.265)	0.2924 ^3^
*p* ^2^		0.0300 ^5^		0.0063 ^5^		
BMC [g]	PRE	2740.52 ± 286.73 (2610.01–2781.04)	2675.00 (2477.00–3005.00)	2929.88 ± 452.03 (2697.47–3162.29)	2879.00 (2684.00–3076.00)	0.1249 ^3^
	POST	2689.81 ± 290.26 (2557.69–2821.93)	2701.00 (2484.00–2929.00)	2897.35 ± 454.86 (2663.49–3131.22)	2792.00 (2605.00–3078.00)	0.0963 ^3^
*p* ^2^		0.0440 ^5^		0.1530 ^5^		
**L1-L4**						
BMD [g/cm^2^]	PRE	1.165 ± 0.151 (1.095–1.233)	1.184 (1.095–1.291)	1.267 ± 0.178 (1.176–1.359)	1.269 (1.146–1.346)	0.0624 ^3^
	POST	1.176 ± 0.157 (1.105–1.248)	1.186 (1.103–1.307)	1.283 ± 0.185 (1.188–1.378)	1.266 (1.135–1.358)	0.0622 ^3^
*p* ^2^		0.0883 ^5^		0.0839 ^5^		
BMC [g]	PRE	63.38 ± 11.59 (58.10–68.65)	66.38 (56.38–70.22)	71.45 ± 15.00 (63.74–79.16)	68.67 (66.28–78.34)	0.0945 ^4^
	POST	63.98 ± 12.02 (58.51–69.46)	67.34 (57.36–73.33)	71.54 ± 14.83 (63.92–79.17)	67.92 (62.51–77.65)	0.0909 ^3^
*p* ^2^		0.3135 ^5^		0.8905 ^5^		
**Femoral neck**						
BMD [g/cm^2^]	PRE	1.017 ± 0.120 (0.962–1.071)	1.013 (0.923–1.080)	1.057 ± 0.155 (0.978–1.137)	1.057 (0.955–1.137)	0.3670 ^3^
	POST	1.006 ± 0.103 (0.959–1.053)	0.978 (0.914–1.065)	1.059 ± 0.141 (0.986–1.131)	1.024 (0.957–1.139)	0.1904 ^3^
*p* ^2^		0.3662 ^6^		0.8951 ^5^		
BMC [g]	PRE	4.81 ± 0.66 (4.51–5.11)	4.78 (4.26–5.08)	5.22 ± 0.99 (4.71–5.73)	4.92 (4.66–6.12)	0.1384 ^3^
	POST	4.61 ± 0.70 (4.29–4.93)	4.63 (4.08–5.07)	5.10 ± 0.88 (4.64–5.55)	4.88 (4.64–5.19)	0.1519 ^4^
*p* ^2^		0.1119 ^5^		0.2663 ^6^		

BMC–bone mineral content, BMD–bone mineral density, END–endurance group, END-ST–endurance-strength group, IQR–interquartile range, PRE–pre-intervention value, POST–post-intervention value, SD–standard deviation, 95% CI–95% of a confidence interval, ^1^
*p*-value for differences between groups, ^2^
*p*-value for differences between pre- and postintervention values, ^3^ unpaired *t*-test, ^4^ Mann–Whitney U-test, ^5^ paired *t*-test and ^6^ Wilcoxon test.

**Table 4 healthcare-10-00821-t004:** The effect of training programmes on body composition.

Variables		END Group (*n* = 21)		END-ST Group (*n* = 17)		*p* ^1^
		Mean ± SD (95% CI)	Median (IQR)	Mean ± SD (95% CI)	Median (IQR)	
**Total body**						
Fat [g]	PRE	41955 ± 7603 (38494–45415)	42329 (36244–48478)	42681 ± 8615 (38252–47110)	42775 (35849–47614)	0.7841 ^3^
	POST	39219 ± 7258 (35915–42522)	37968 (34728–46281)	40025 ± 8442 (35684–44366)	38833 (32928–45588)	0.7534 ^3^
*p* ^2^		<0.0001 ^5^		<0.0001 ^5^		
Lean [g]	PRE	45969 ± 5468 (43480–48458)	44017 (42118–50221)	47844 ± 6280 (44615–51073)	45605 (44368–51138)	0.3319 ^3^
	POST	46598 ± 5961 (43884–49311)	44881 (42854–48810)	48640 ± (45354–51926)	46895 (44192–51581)	0.3160 ^3^
*p* ^2^		0.1096 ^5^		0.0005 ^5^		
**Male area (android)**						
Fat [g]	PRE	4244 ± 990 (3793–4694)	4336 (3465–4757)	4278 ± 1002 (3763–4793)	4038 (3464–4920)	0.9161 ^3^
	POST	3906 ± 960 (3469–4343)	3701 (3319–4316)	3956 ±1090 (3396–4516)	3660 (3083–4685)	0.9065 ^4^
*p* ^2^		<0.0001 ^6^		0.0003 ^5^		
Lean [g]	PRE	3630 ± 681 (3320–3940)	3301 (3146–4196)	3815 ± 833 (3387–4243)	3473 (3263–4236)	0.4811 ^4^
	POST	3478 ± 646 (3184–3772)	3247 (3111–3727)	3683 ± 775 (3284–4081)	3321 (3144–4258)	0.4453 ^4^
*p* ^2^		0.0117 ^6^		0.0975 ^6^		
**Female area (gynoid)**						
Fat [g]	PRE	6936 ± 1425 (6287–7584)	7342 (5821–7708)	7263 ± 1715 (6382–8145)	7538 (6035–8051)	0.5237 ^3^
	POST	6412 ± 1315 (5814–7011)	6701 (5482–6938)	6563 ± 1617 (5732–7395)	6609 (5522–7124)	1.0000 ^4^
*p* ^2^		<0.0001 ^5^		<0.0001 ^5^		
Lean [g]	PRE	6785 ± 965 (6346–7224)	6575 (6060–7288)	6915 ± 950 (6426–7403)	6559 (6299–7274)	0.6799 ^3^
	POST	7054 ± 1008 (6595–7513)	6844 (6572–7655)	7529 ± 1106 (6960–8098)	7138 (6694–8253)	0.1750 ^3^
*p* ^2^		0.0013 ^5^		0.0001 ^5^		
**Arms**						
Fat [g]	PRE	3571 ± 721 (3243–3899)	3660 (3060–4181)	3514 ± 569 (3221–3806)	3465 (2973–3905)	0.7900 ^3^
	POST	3506 ± 604 (3231–3781)	3503 (3002–4038)	3313 ± 648 (2980–3647)	3212 (2794–3754)	0.3497 ^3^
*p* ^2^		0.4178 ^5^		0.0442 ^6^		
Lean [g]	PRE	4330 ± 746 (3991–4670)	4597 (3973–4838)	4405 ± 843 (3972–4839)	4493 (3826–4885)	0.7728 ^3^
	POST	4312 ± 540 (4066–4557)	4444 (3922–4637)	4284 ± 757 (3894–4673)	4372 (3819–4555)	0.8953 ^3^
*p* ^2^		0.8400 ^5^		0.3094 ^5^		
**Legs**						
Fat [g]	PRE	14136 ± 3663 (12469–15804)	14647 (11214–16032)	13718 ± 3761 (11784- 15651)	14691 (11252–16091)	0.7314 ^3^
	POST	13242 ± 3638 (11585–14898)	14055 (10468–14925)	12992 ± 3635 (11123–14861)	13462 (11597–15094)	0.8347 ^3^
*p* ^2^		0.0002 ^5^		0.0006 ^5^		
Lean [g]	PRE	15067 ± 2240 (14047–16086)	14821 (13316–16491)	15263 ± 1594 (14444–16083)	15311 (13821–16059)	0.7626 ^3^
	POST	15584 ± 2690 (14357–16807)	15272 (13324–16917)	16442 ± 2293 (15262–17621)	16245 (14308–17815)	0.3030 ^3^
*p* ^2^		0.0129 ^5^		0.0003 ^6^		
**Trunk**						
Fat [g]	PRE	23013 ± 4626 (20907–25119)	23297 (19724–25380)	24485 ± 5498 (21658–27312)	23574 (20840–26572)	0.3758 ^3^
	POST	21512 ± 4430 (19496–23529)	21840 (18644–23035)	22790 ± 5203 (20115–25465)	21847 (19603–25749)	0.4188 ^3^
*p* ^2^		0.0011 ^5^		0.0014 ^6^		
Lean [g]	PRE	23522 ± 3654 (21858–25185)	22540 (20780–25583)	25097 ± 5062 (22495–27700)	22415 (21872–27977)	0.5972 ^4^
	POST	23613 ± 3691 (21933–25293)	22752 (21423–24736)	24792 ± 4431 (22514–27070)	22090 (21739–27338)	0.5376 ^4^
*p* ^2^		0.7480 ^5^		0.5228 ^6^		

END–endurance group, END-ST–endurance-strength group, IQR–interquartile range, PRE–pre-intervention value, POST–post-intervention value, SD–standard deviation, 95% CI–95% confidence interval, ^1^
*p*-value for differences between groups, ^2^
*p*-value for differences between pre- and postintervention values, ^3^ unpaired *t*-test, ^4^ Mann–Whitney U-test, ^5^ paired *t*-test and ^6^ Wilcoxon test.

**Table 5 healthcare-10-00821-t005:** Comparison of the differences (Δ) in the effect of endurance and endurance-strength training on bone turnover marker levels.

Variables	END Group (*n* = 21)		END-ST Group (*n* = 17)		*p* ^1^
	Mean ± SD (95% CI)	Median (IQR)	Mean ± SD (95% CI)	Median (IQR)	
Δ BAP [U/L]	0.51 ± 6.03 (−2.12–3.39)	1.18 (−2.28–2.52)	−0.87 ± 7.17 (−2.94–1.37)	0.16 (−2.98–1.56)	0.3341
Δ CTX-1 [ng/mL]	0.01 ± 0.13 (−0.06–0.08)	0.01 (−0.08–0.11)	0.05 ± 0.11 (−0.04–0.014)	0.09 (−0.06–0.18)	0.3381
Δ OC [ng/mL]	0.74 ± 2.31 (−0.18–1.74)	0.79 (−0.25–2.35)	0.85 ± 2.38 (−0.14–1.94)	0.72 (−0.36–1.41)	0.4717
Δ TRAP 5b [U/L]	0.27 ± 0.67 (−0.10–0.60)	0.260 (−0.04–0.80)	0.67 ± 0.68 (0.31–1.00)	0.57 (0.14–0.79)	0.2018

BAP–bone alkaline phosphatase, CTX-1–α-collagen type 1 cross-linked C-terminal telopeptide, END–endurance group, END-ST–endurance-strength group, IQR–interquartile range, OC–osteocalcin, SD–standard deviation, TRAP 5b–tartrate-resistant acid phosphatase serum band 5, 95% CI–95% confidence interval, Δ–changes (post- minus preintervention values), ^1^
*p*-value for differences between groups and the ANCOVA test, adjusted for the baseline measures as a covariate.

**Table 6 healthcare-10-00821-t006:** Comparison of the differences (Δ) in the effect of endurance and endurance-strength training on bone parameters.

Variables	END Group (*n* = 21)		END-ST Group (*n* = 17)		*p* ^1^
	Mean ± SD (95% CI)	Median (IQR)	Mean ± SD (95% CI)	Median (IQR)	
**Total body**					
Δ BMD [g/cm^2^]	−0.019 ± −0.038 (−0.032–−0.005)	−0.021 (−0.033–−0.004)	−0.019 ± −0.044 (−0.031–−0.007)	−0.022 (−0.032–0.004)	0.9923
Δ BMC [g]	−47.20 ± 235.84 (−94.38–−1.07)	−44.00 (−104.00–26.00)	−30.83 ± 249.46 (−77.59–14.95)	−46.00 (−93.00–49.00)	0.9020
**L1-L4**					
Δ BMD [g/cm^2^]	0.012 ± 0.017 (−0.001–0.025)	0.011 (−0.005–0.029)	0.016 ± 0.010 (−0.002–0.034)	0.014 (−0.001–0.052)	0.8416
Δ BMC [g]	0.55 ± 3.36 (−0.66–1.79)	0.90 (−0.91–2.42)	0.02 ± 3.33 (−1.35–1.43)	−0.08 (−1.15–1.52)	0.6308
**Femoral neck**					
Δ BMD [g/cm^2^]	−0.009 ± 0.064 (0.024–0.006)	−0.004 (−0.023–0.010)	0.004 ± 0.054 (−0.018–0024)	0.002 (−0.033–0.027)	0.1083
Δ BMC [g]	−0.112 ± 0.725 (−0.329–0.077)	−0.080 (−0.450–0.050)	−0.087 ± 0.995 (−0.241–0.053)	0.000 (−0.210–0.050)	0.3787

BMC–bone mineral content, BMD–bone mineral density, END–endurance group, END-ST–endurance-strength group, IQR–interquartile range, SD–standard deviation, 95% CI–95% confidence interval, Δ–changes (post- minus preintervention values), ^1^
*p*-value for differences between groups and the ANCOVA test, adjusted for the baseline measures as a covariate.

**Table 7 healthcare-10-00821-t007:** Comparison of the differences (Δ) in the effect of endurance and endurance-strength training on body composition.

Variables	END Group (*n* = 21)		END-ST Group (*n* = 17)		*p* ^1^
	Mean ± SD (95% CI)	Median (IQR)	Mean ± SD (95% CI)	Median (IQR)	
**Total body**					
Δ fat [g]	−2802 ± 5329 (−3607–−1975)	−2226 (−3284–−1812)	−2716 ± 5246 (−3673–−1727)	−2571 (−3951–−1883)	0.8324
Δ lean [g]	608 ± 1927 (−180–1403)	685 (82–1192)	793 ± 2848 (404–1184)	745 (−4–1483)	0.7496
**Male (android)**					
Δ fat [g]	−334 ± 879 (−457–−212)	−233 (−357–−178)	−320 ± 853 (−471–−169)	−376 (−478–−275)	0.8638
Δ lean [g]	−138 ± 696 (−252–−28)	−81 (−225–−37)	−110 ± 635 (−263–36)	−152 (−222–27)	0.5828
**Female (gynoid)**					
Δ fat [g]	−523 ±1317 (−673–−373)	−507 (−773–−304)	−700 ± 1230 (−914–−486)	−637 (−1004–−439)	0.2027
Δ lean [g]	227 ± 224 (71–400)	227 (32–504)	568 ± 188 (325–836)	510 (331–829)	0.0211
**Arms**					
Δ fat [g]	−96 ± 795 (−258–80)	−122 (−185–53)	−258 ± 730 (−476–−8)	−295 (−480–−162)	0.1746
Δ lean [g]	−45 ± 839 (−227–148)	−137 (−328–222)	−169 ± 777 (−409–96)	−154 (−400–−7)	0.4530
**Legs**					
Δ fat [g]	−879 ± 2563 (−1290–−472)	−948 (−1123–−570)	−719 ±−2779 (−1079 - −362)	−666 (−1229–−2779)	0.5938
Δ lean [g]	454 ± 968 (45–886)	354 (94–1061)	1141 ± 1012 (674–1630)	774 (583–1567)	0.0381
**Trunk**					
Δ fat [g]	−1671 ± 4321 (−2515–−761)	−1468 (−2113–−862)	−1784 ± 4590 (−2521–−996)	−1990 (−2680–−1512)	0.9346
Δ lean [g]	90 ± 2377 (−493–674)	190 (−648–553)	−307 ± 2383 (−963–350)	−27 (−639–378)	0.5387

END–endurance group, END-ST–endurance-strength group, IQR–interquartile range, SD–standard deviation, 95% CI–95% confidence interval, Δ–changes (post- minus preintervention values), ^1^
*p*-value for differences between groups and the ANCOVA test, adjusted for the baseline measures as a covariate.

## Data Availability

The data presented in this study are available on request from the corresponding author (E.M.). The data are not publicly available due to the disagreement of the study participants.

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
