# Peer review of "The Effect of Endurance and Endurance-Strength Training on Bone Health and Body Composition in Centrally Obese Women—A Randomised Pilot Trial"

_healthcare, 2022, doi:10.3390/healthcare10050821_

Round 1

Reviewer 1 Report

Title too long, “abdominally obese” seems a particular unusual situation and it is not defined in text. I find the same text in author’s other publications cited here. I recommend rephrasing this

Abstract well structured

Introduction presents definition of obesity and data from literature analyzing factors influencing obesity

Authors provide data regarding ethical approval and patients informed consent.

Inclusion and exclusion criteria are briefly presented. These criteria should be presented in detail in the present study, in text or table. One cannot refer to inclusion exclusion criteria and sent the reader to another study.

Primary and secondary purpose of the study are presented.

Training programs are presented in detail and reproducible.

Protocols for data registration are presented for every parameter.

Row 224 ­- The study was conducted between January and March 2013. This information should be presented in Materials and Methods section

Row 232 - The baseline characteristics of the study population were also published previously (19-22). Again, the reader is sent to other publications for the same population

Same groups, same patients, and same data from table 1 are already published in paper cited at number 20 (also table 1).

The information presented in tables 2-7 is relevant for the results of this study, but the tables are with so many data that are difficult to read and the important information is diluted.

Discussion section starts with presentation of the most important results. Results are compared with other finding in literature.

The limitations of this study are presented

The title refers to this study as “three-month randomized pilot trial”. Patients were recruited in 2013. Evaluation for only 3 months of BMD and BMC is not relevant for a proper evaluation on medium and long term of any kind of treatment, considering the bone turnover. My simple deduction is that either the authors weren’t able (for various reasons) to continue the evaluation of those training programs, or the programs have no effect on medium to long term follow up. Authors have already published in Healthcare a study with patients recruited in 2016 (doi: 10.3390/healthcare9081074) analyzing The Effect of Endurance and Endurance-Strength Training on BMD in Abdominally Obese Women. This study is presented as subsequent, with a new training program - reference 24.

Conclusions are related to this study.

Reviewer 2 Report

Evidence between physical exercise and bone health should be indicated in the introduction. Indicate if they meet the training criteria of this study in order to obtain improvements in bone health.

Explain the sample in more detail. It would be important to determine if they are sedentary or physically active people. Whether or not they are menopausal.

It is necessary to identify who designs the training. In resistance training, indicate how the intensity of the workload is adjusted over the three months. Load from 50 to 80 are indicated but it is necessary to indicate their progression.

Regarding endurance-strength training, it is necessary to explain that it meets the requirements to be a training with those characteristics. The protocol of strength work (number of sets, repetitions) should be explained.

To work on strength, it is necessary to calculate a 1Repetition Maximum. To be able to establish volume and intensity for each muscle group. This is a relevant aspect to talk about resistance-strength training.

Round 2

Reviewer 1 Report

Dear authors,

Thank you for your effort, this is a good paper providing relevant scientific data.

Reviewer 2 Report

The manuscript has improved. But strength training has weaknesses. It would be advisable to indicate this in the limitations.
